# ITERATIVE DECODING FOR COMPOSITIONAL GENERALIZATION IN TRANSFORMERS

## ABSTRACT

Deep learning models do well at generalizing to in-distribution data but struggle to generalize compositionally, i.e., to combine a set of learned primitives to solve more complex tasks. In particular, in sequence-to-sequence (seq2seq) learning, transformers are often unable to predict correct outputs for even marginally longer examples than those seen during training. This paper introduces *iterative decoding*, an alternative to seq2seq learning that (i) improves transformer compositional generalization and (ii) evidences that, in general, seq2seq transformers do not learn iterations that are not unrolled. Inspired by the idea of compositionality— that complex tasks can be solved by composing basic primitives—training examples are broken down into a sequence of intermediate steps that the transformer then learns iteratively. At inference time, the intermediate outputs are fed back to the transformer as intermediate inputs until an end-of-iteration token is predicted. Through numerical experiments, we show that transfomers trained via iterative decoding outperform their seq2seq counterparts on the PCFG dataset, and solve the problem of calculating Cartesian products between vectors longer than those seen during training with 100% accuracy, a task at which seq2seq models have been shown to fail. We also illustrate a limitation of iterative decoding, specifically, that it can make sorting harder to learn on the CFQ dataset.

## 1 INTRODUCTION

Deep learning architectures achieve state-of-the-art results in a wide array of machine learning problems, where their impressive performance is attributed to their ability to generalize (Goodfellow et al., 2016; LeCun et al., 2015). However, this ability is typically limited to generalization under the statistical learning paradigm, i.e., in-distribution generalization, and does not encompass generalizing compositionally. Compositional generalization is the ability of a model to combine a set of learned primitives to execute more complex tasks. For instance, for a ground robot whose motion planner has learned to execute the instructions "walk", "jump", and "jump right", generalizing compositionally would be to be able to execute the instruction "walk right" (Lake & Baroni, 2018).

In machine learning models, compositional generalization is desirable for two reasons. First, because it is a crucial aspect of intelligence observed in both humans and classical artificial intelligence techniques. In humans, a prevailing example is the way children can solve complicated mathematical expressions after being taught basic arithmetics. Second, because it can increase a model's data efficiency. By endowing models with the ability to extrapolate to unseen examples that are more complex, or complex in different ways than seen at training, compositionality acts as an implicit mechanism for data augmentation.

In this paper, our goal is to increase compositional generalization in transformers, with particular focus on natural language-like tasks—where compositionality is key. Understanding that for models to be able to execute composite tasks they need to be taught *how to compose*, we introduce *iterative decoding*, an alternative to sequence-to-sequence (seq2seq) learning that decomposes the process of mapping the inputs to the outputs of each example into a sequence of intermediate steps which the transformer learns to perform iteratively. During training, each input-output pair is converted into a sequence of "intermediate input-intermediate output" pairs, which are task specific. During prediction, the intermediate outputs predicted by the transformer are adapted into the subsequent intermediate inputs, which are fed back to the transformer until an end-of-iteration token is produced.

Our main contributions are (i) showing that iterative decoding, especially when combined with other architectural modifications such as relative attention (Shaw et al., 2018) and copy decoders (Ontañón et al., 2021), largely improves compositional generalization in transformers, and (ii) evidencing that, in general, seq2seq transformers cannot learn iterations unless they are unrolled. We show this through numerical experiments on two compositionally hard splits of PCFG (Hupkes et al., 2020), a string editting dataset; and on a cartesian product dataset, where the goal is to generalize to longer input vectors than those on which the model is trained. We also present numerical results on CFQ (Keysers et al., 2019), a semantic parsing dataset consisting of natural language questions paired with SPARQL queries. Importantly, the results obtained in this dataset evidence a limitation of iterative decoding, which is that it can be sensitive to the ordering of the intermediate steps.

The rest of this paper is organized as follows. Section 2 discusses related work on compositional generalization, transformers and on the PCFG, cartesian product, and CFQ datasets. Section 3 introduces iterative decoding. Iterative decoding strategies specific to each dataset and numerical results are presented and discussed in Section 4. Section 5 presents future research directions and concluding remarks.

## 2 BACKGROUND

In this section, we introduce some background and related work on compositional generalization, transformer architectures and on the PCFG, Cartesian product and CFQ datasets, used in our experiments.

### 2.1 COMPOSITIONAL GENERALIZATION

Compositional generalization (or compositionality) refers to the ability of a model that has learned to perform a set of basic operations—*primitives*—to generalize to more complex operations, i.e., operations consisting of *compositions* of the learned primitives (Lake & Baroni, 2018). Examples of operations requiring compositionality are shown in Figure 1 for three datasets. For instance, the top-left corner shows an example from the PCFG (Hupkes et al., 2020) dataset. In some versions of this dataset, the model is trained to solve several atomic string editing operations (such as `copy` and `swap_first_last`), and how to compose them. During testing, the model is tested by asking it to compose a longer number of operations than seen during training. Hence, the model has to be able to generalize compositionally. This string editing example can be seen as an instance of *productivity*, one of the five types of compositional generalization identified by Hupkes et al. (2020) which involves generalizing to longer examples than those seen during training. Another type of compositional generalization is *systematicity*, the ability to recombine known parts and rules in ways different than those seen during training.

Early works on compositionality have explored the limitations of different machine learning models in generalizing compositionally. Liška et al. (2018) showed that, while it is theoretically possible for a recurrent neural network (RNN) to generalize in this way, only a small fraction of the models they trained behaved compositionally. Lake & Baroni (2018) proposed SCAN, a dataset consisting of navigation commands to be mapped to action sequences, and observed that while RNNs trained on it only generalized well when the differences between the training and test sets where small, they failed when more systematic compositional skills were required. Other datasets created with the purpose of measuring compositionality include PCFG (Hupkes et al., 2020) and CFQ (Keysers et al., 2019), where both long-short term memory (LSTM) and transformer-based architectures have been shown to perform poorly.

More recently, a popular research direction is to try to endow these machine learning models with a "compositional generalization bias". Kim et al. (2021) saw benefits in converting CFQ into a classification task and using structural annotations (e.g., entity links) as attention masks in transformers. Ontañón et al. (2021) were able to improve transformer compositional generalization on a variety of compositionally hard datasets by making architectural modifications such as relative attention, copy decoders, and weight sharing. Taking a similar approach, Csordás et al. (2021) observed performance improvements from relative position encodings and scaled embeddings. Other strategies to improve compositional generalization include increased pretraining (Furrer et al., 2020), data augmentation (Andreas, 2019) and differentiable neurocomputers (Graves et al., 2016).

```
PCFG

IN: copy swap_first_last E18 B11 T13 W20

OUT: W20 B11 T13 E18
```

```
Cartesian product

IN: 1 4 3 [SEP] a d

OUT: 1 a 1 d 4 a 4 d 3 a 3 d
```

```
CFQ

IN: Did a person marry a cinematographer ,
influence M1 , and influence M2

OUT: SELECT count(*) WHERE {
 ?x0 a ns:people.person .
 ?x0 ns:influence.influence_node.influenced M1 .
 ?x0 ns:influence.influence_node.influenced M2 .
 ?x0 ns:people.person.spouse_s ?x1 .
 ?x1 a ns:film.cinematographer .
 FILTER ( ?x0 != ?x1 ) }
```

Figure 1: Examples of input-output pairs in the PCFG, cartesian product, and CFQ datasets.

Closely related to our work, PonderNet (Banino et al., 2021) trains a model that iterates internally to achieve a better compromise between training accuracy and generalization. It predicts both an output and a halting probability at each step, operating recurrently. Iterative decoding also operates recurrently, but two important differences are that (i) in iterative decoding the intermediate steps are supervised and interpretable, and (ii) rather than predicting a halting probability at each step, iterative decoding trains the model to produce a special token to indicate end of iteration.

## 2.2 TRANSFORMER MODEL

In this paper we focus on transformer models. Despite struggling to generalize compositionally, transformer-based architectures such as BERT (Devlin et al., 2018) and T5 (Raffel et al., 2019) were popularized by their remarkable performance in machine translation (Zhu et al., 2020), question answering (Ainslie et al., 2020), summarization (Zhang et al., 2019) and other natural language processing (NLP) tasks.

Introduced by Vaswani et al. (2017), the basic transformer model is composed of an encoder and a decoder. The encoder is made up of layers consisting of a self-attention sublayer and a feedforward sublayer. The decoder has the same structure, but with an additional attention sublayer to compute the decoder-to-encoder attention. The input to the transformer is a sequence of token embeddings. Since these embeddings do not carry information about the position of each token in the sequence, a position encoding is typically added to the input embeddings. These are then fed to the encoder, which encodes all tokens at once and forwards the result to the decoder. From the encoded input and the decoded output tokens generated so far, the decoder generates the distribution of the next output token, one token at a time.

We experiment with two extensions to the original transformer architecture that were previously shown to improve compositional generalization (Ontañón et al., 2021): relative position encodings and copy decoding. To each pair of tokens in the input, relative position encodings assign a label that is equal to the minimum between their relative distance and a fixed relative attention radius. An important characteristic of relative position encodings is that they are position invariant, which means that two tokens that are $k$ positions apart will attend to each other in the same way regardless of their absolute positions in the sequence. Copy decoding involves adding a learnable parameter that allows to switch between the decoder and a copy decoder which produces an independent embedding that can be interpreted as a "copy" from the input sequence.

## 2.3 DATASETS

We consider three datasets—PCFG, cartesian product, and CFQ—which are discussed in detail in the following sections.

### 2.3.1 PCFG

PCFG is an artificial translation dataset proposed by Hupkes et al. (2020) and generated by a probabilistic context free grammar. The input data consists of string editing instructions with four types of

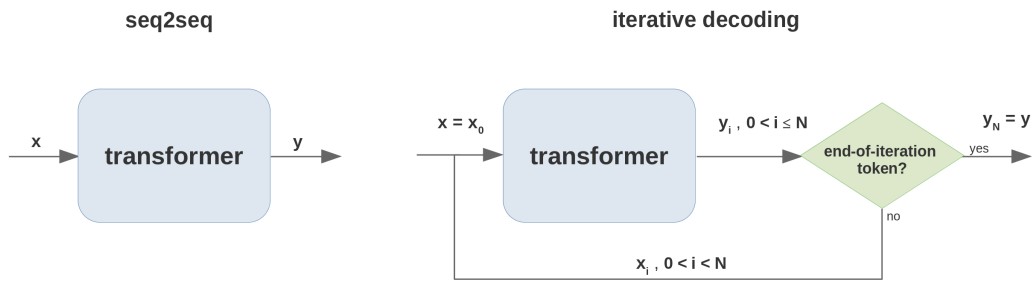

Figure 2: Prediction routines for the seq2seq transformer (left) and iterative decoding transformer (right).

tokens: unary operation tokens (e.g., `reverse`), binary operation tokens (e.g., `append`), a string separation token "`,`" (to separate arguments of binary operations), and string elements (e.g., `B10`, `D2`). The output data consists of the strings resulting from the application of the operations; see the top left corner of Figure 1 for an example.

There are six training-test splits of the PCFG dataset. The first is a random split which we use as a baseline. The other five are compositionally hard splits used to measure the five different types compositional generalization. We focus on two of them: productivity and systematicity. In the productivity split, the training samples have up to 8 string operations, while the test samples have 9 or more. In the systematicity split, there is no restriction on the number of operations in either set, but in the test set they are combined in different ways than in the training set.

The main challenge of the PCFG dataset is that it requires learning ten string editing operations, some of which are very similar. The unary operation `echo`, for instance, only differs from `copy` by repeating the last element of the string. While transformers generally achieve good performance in the random split of the PCFG dataset, the productivity and systematicity splits are harder because transformers tend to learn shortcuts (instead of learning the mechanics of each operation). In particular, a key difficulty of the productivity split is that the model needs to learn to do "recursion" and apply an arbitrary number of operations when input examples grow in length. In some cases, the strings to modify can also be very long, which places an additional capacity burden on transformers by requiring them to learn how to copy strings.

### 2.3.2 CARTESIAN PRODUCT

In the cartesian product dataset, the inputs are two vectors and the outputs are their cartesian product. The first input vector is a vector of numbers. The second is a vector of letters, separated from the numbers by the special token `[SEP]`. Both numbers and letters are picked at random, without repetition, from the decimal digits and the first ten letters of the alphabet respectively. A possible input-output pair is shown on the bottom left corner of Figure 1.

We consider four splits of the cartesian product dataset. In the first split, both the training and test set consists of samples with up to five numbers and letters drawn i.i.d and split at random. This is the "easy split". In the other splits, the training set is the same as in the first split, but the test set consists of examples with six numbers and five letters for the second split, five numbers and six letters for the third, and six numbers and letters for the fourth respectively. These are "hard splits", which we use to measure productivity. The productivity splits of the cartesian product dataset are remarkably hard; even transformers with some compositional generalization ability in other mathematical datasets have been seen to fail (Ontañón et al., 2021). This is due to the fact that, in order to solve cartesian products, models need to learn to execute two nested loops. Moreover, the output is quadratic on the size of the inputs. For models that have to learn to predict an end-of-sequence token, extrapolating to longer sequences than those seen during training has been shown to be difficult (Newman et al., 2020).

**PCFG**

```
swap_first_last repeat copy J4 A9 N7 V8

swap_first_last repeat J4 A9 N7 V8

swap_first_last J4 A9 N7 V8 J4 A9 N7 V8

V8 A9 N7 V8 J4 A9 N7 J4 [END]
```

**Cartesian product (row)**

```
IN: 7 3 8 [SEP] c a                    OUT: 7 c 7 a
IN: 7 3 8 [SEP] c a [SEP2] 7 c 7 a      OUT: 3 c 3 a
IN: 7 3 8 [SEP] c a [SEP2] 3 c 3 a      OUT: 8 c 8 a [END]
```

**Cartesian product (token)**

```
IN: 2 5 [SEP] e f                      OUT: 2 e
IN: 2 5 [SEP] e f [SEP2] 2 e           OUT: 2 f
IN: 2 5 [SEP] e f [SEP2] 2 f           OUT: 5 e
IN: 2 5 [SEP] e f [SEP2] 5 e           OUT: 5 f [END]
```

**CFQ**

```
IN: Did a person marry a cinematographer , influence M1     OUT: SELECT count(*) WHERE {
IN: Did a person marry a cinematographer , influence M1     OUT: ?x0 a ns:people.person .
    [SEP2] SELECT count(*) WHERE {
IN: Did a person marry a cinematographer , influence M1     OUT: ?x0 ns:influence.influence_node.influenced M1 .
    [SEP2] SELECT count(*) WHERE { ?x0 a ns:people.person .

                    ...                                                 ...

IN: Did a person marry a cinematographer , influence M1     OUT: FILTER ( ?x0 != ?x1 ) } [END]
    [SEP2] SELECT count(*) WHERE { ?x0 a ns:people.person .
    ?x0 ns:influence.influence_node.influenced M1 . ?x0
    ns:people.person.spouse_s ?x1 . ?x1 a
    ns:film.cinematographer
```

Figure 3: Examples of intermediate input-output pairs in the PCFG and cartesian product datasets.

### 2.3.3   CFQ

Introduced by Keysers et al. (2019), the CFQ dataset consists of natural language questions and their corresponding SPARQL queries against the Freebase knowledge base. Hence, it can be used to perform semantic parsing by taking the questions as the inputs and the queries as the outputs. As detailed in (Keysers et al., 2019), compositionally hard splits of the CFQ dataset can be generated by maximizing its compound divergence and minimizing its atom divergence. In this paper, we focus on the MCD1 split of the CFQ dataset. An example of question and query from this split are shown on the right hand side of Figure 1.

One of the difficulties of the CFQ dataset is that some of its examples require solving cartesian products. As such, CFQ may face similar challenges to the ones described in the previous section. Another difficulty is associated with the ordering of the clauses in the SPARQL query, which are ordered alphabetically by convention. Not only is this ordering different than the one implied by the question, it also requires transformers to learn how to sort.

## 3   ITERATIVE DECODING

To improve compositional generalization in transformers, we introduce iterative decoding. As illustrated on the right hand side of Figure 2, iterative decoding consists of predicting a series of intermediate outputs $y_1, y_2, y_3, \ldots$ from an input $x = x_0$, and then adapting these outputs into intermediate inputs $x_i = y_i$, $i > 0$, that are fed back to the model until the final output $y_N = y$ is predicted. This can be visualized by considering the PCFG example $x = $ "`swap_first_last repeat copy J4 A9 N7 V8`" on the left hand side of Figure 3. A seq2seq transformer trained on the PCFG dataset is expected to output $y = $ "`V8 A9 N7 V8 J4 A9 N7 J4 [END]`" in one forward pass (i.e, to go from top to bottom in the figure). However, in iterative decoding, the transformer's output to the input $x_0 = x$ would be $y_1 = $ "`swap_first_last repeat J4 A9 N7 V8`", which is the first intermediate output of iterative decoding (the second string from the top), and corresponds to just executing one of the operations in the input, `copy`. Setting $x_1 = y_1$ and feeding this instruction back to the transformer, we obtain $y_2 = $ "`swap_first_last J4 A9 N7 V8 J4 A9 N7 V8`" (the third string from the top). The intermediate output $y_2$ then becomes the intermediate input $x_2$, which the transformer processes to produce the final output $y_3 = $ "`V8 A9 N7 V8 J4 A9 N7 J4 [END]`".

The main motivation for iterative decoding comes from the very idea of compositional generalization—by decomposing complex instructions into intermediate steps, iterative decoding essentially teaches models how to compose. Another motivation, related to the first, is that iterative decoding mimics how humans are taught how to perform many compositional tasks. For example, when teaching how to solve the arithmetic expression $2 \times (1 + 1)$, first we demonstrate how to solve the inner sum, then how to eliminate the parentheses, and finally how to compute the product. A third motivation for iterative decoding is that learning step by step can potentially prevent shortcut learning, one of the biggest obstacles to compositional generalization in seq2seq, because it reduces the difficulty of the tasks that the transformer needs to learn to execute in a single forward pass.

To implement iterative decoding, we need to modify both how models are trained and how they predict. The most important change in training is that, instead of being trained on the original inputs and outputs, iterative decoding transformers are trained on the "intermediate input-intermediate output" pairs $(x_{i-1}, y_i)$ for $1 \leq i \leq N$. These intermediate inputs and outputs are pre-generated, and their form is specific to each task (see Section 4 for examples for the PCFG, cartesian product and CFQ datasets). Additionally, an end-of-iteration token has to be added to the training outputs so that the model learns when to stop. Intuitively, the intermediate outputs correspond to providing supervision to the transformers on which are the steps that it should execute to solve the task, similar to how humans are taught how to perform mathematical operations step by step, rather then learning from input-output mappings of complex expressions.

Depending on the number of intermediate steps necessary to iteratively decode a given example, the prediction requires multiple forward passes of the transformer. Hence, it is implemented as a while loop where the stopping condition is finding the end-of-iteration token. This is illustrated on Figure 2, which compares seq2seq (left) with iterative decoding predictions (right). After each intermediate prediction, in this paper we use some data processing step to adapt the intermediate outputs into the following intermediate inputs in some datasets. As we could see from the example above, this is not necessary for the PCFG dataset, because it has a built-in recursive structure. But it will be for the cartesian product and CFQ datasets as we detail in Section 4. While, ideally this processing step should be learned, we provide it manually in this paper to being to understand the possibilities of iterative decoding.

## 4  RESULTS AND DISCUSSION

In this section we describe the iterative decoding schemes for the PCFG, cartesian product and CFQ datasets, and present and discuss numerical results obtained for seq2seq and iterative decoding transformers. All transformers have $\ell = 6$ encoder/decoder layers, embedding dimension $d = 64$, feedforward dimension $f = 256$ and $h = 4$ attention heads. For each dataset, each experiment is repeated 3 times. Additional implementation details can be found in Appendix A.

### 4.1  PCFG

We apply iterative decoding to the PCFG dataset by breaking down each example into a number of intermediate steps equal to the number of string editing operations present in the original input. Each intermediate step solves the rightmost instruction in the current intermediate input. Up until the last step, all of the intermediate outputs are themselves string editing instructions. Hence, the intermediate outputs do not need to be adapted and serve as the intermediate inputs to the next step. Hence, the only additional processing of the dataset needed for iterative decoding is the addition of the end-of-iteration token `[END]` at the end of the final output. An iterative decoding PCFG example with three intermediate steps is shown on the left hand side of Figure 3.

To compare iterative decoding with seq2seq, we start by considering a basic transformer model with absolute position encodings as in the original architecture by Vaswani et al. (2017). This is so we can observe the advantages of iterative decoding in the absence of any other compositional generalization biases. The sentence level-accuracy achieved by these models on the training and test sets of the random, productivity and systematicity splits of the PCFG dataset is shown in the first two rows of Table 1. From the first row, we see that in the random split of the data the test accuracy of the seq2seq model is close to its training accuracy. This indicates that the model generalizes well to in-distribution samples. In contrast, in the productivity and systematicity splits there is a

Table 1: Sentence accuracy on the training and test sets for the random, productivity and systematicity splits of the PCFG dataset.

| Model | Random | | Productivity | | Systematicity | |
|---|---|---|---|---|---|---|
| | train | test | train | test | train | test |
| seq2seq-abs | 87.2% | 85.9% | 90.6% | 34.2% | 89.5% | 64.8% |
| it-dec-abs | 96.5% | 93.2% | 96.6% | 45.7% | 94.6% | 82.9% |
| seq2seq-rel | 98.1% | 97.4% | 98.8% | 65.1% | 98.2% | 85.7% |
| it-dec-rel | 99.8% | 99.2% | 100% | 91.9% | 99.7% | 97.0% |
| seq2seq-rel&copy | 97.7% | 97.0% | 98.3% | 63.9% | 98.2% | 85.1% |
| it-dec-rel&copy | 99.7% | **99.4%** | 100% | **93.3%** | 99.8% | **97.8%** |

dramatic drop in test accuracy. This shows that the basic seq2seq transformer struggles to generalize compositionally. In the second row of Table 1, we see that for the iterative decoding transformer the gap between training and test accuracy is much smaller, especially in the systematicity split. This indicates that iterative decoding increases the ability of transformers to generalize compositionally.

Although iterative decoding helps with compositionality, the test accuracy achieved by the iterative decoding transformers is still low compared to their training accuracy. This implies that composing individual operations into complex instructions is only one facet of compositionality, which makes sense as decomposing complex instructions into individual operations only helps if the model can execute each operation correctly (see Appendix B for more details). To verify this claim numerically, we repeat our experiments with transformers including modifications shown to increase compositional generalization in seq2seq learning (Ontañón et al., 2021). These modifications are: (a) replacing absolute attention by *relative attention* (Shaw et al., 2018), and (b) adding a learnable parameter that allows to switch between the decoder and a *copy decoder* (Ontañón et al., 2021).

The results for seq2seq and iterative decoding with relative attention are shown in the third and fourth rows of Table 1. The relative attention radius is $r = 8$. As expected, relative attention helps both models with compositionality, particularly in the productivity split. However, iterative decoding achieves a much better test accuracy, of over 90%, on both the productivity and systematicity splits. Results for transformers with relative attention and copy decoders are shown in the fifth and sixth rows of Table 1. While adding a copy decoder does not improve compositional generalization in the seq2seq transformer, it helps in iterative decoding, nearly closing the gap between training and test accuracy on the systematicity split, and leading to a 2% increase in test accuracy on the productivity split—which can be attributed to the ability to copy the longer strings in this split.

## 4.2 CARTESIAN PRODUCT

To iteratively decode a cartesian product, we first need to define what are going to be the iterative decoding intermediate steps. There are many possibilities; we could, for instance, define an intermediate step as predicting one output token at a time, or a sequence of tokens with fixed length. In this paper we consider two options. The first is decoding one *row* at a time. As illustrated on the top right corner of Figure 3, this entails decoding the cartesian product between one element from the first vector and all elements of the second vector at each intermediate step. The second is decoding one *token pair* at a time. As illustrated on the bottom right corner of Figure 3, this entails decoding only the product between one element of the first vector and one element of the second vector at each intermediate step. If the lengths of the input vectors are $\ell_1$ and $\ell_2$ respectively, decoding row by row requires $\ell_1$ and token by token $\ell_1 \times \ell_2$ intermediate steps.

When decoding one row at a time, the intermediate output at a given step is the current row. Similarly, when decoding one pair of tokens at a time, the intermediate outputs are the current token pairs. Since these intermediate outputs do not carry any information about the next row or pair of tokens to predict, they cannot be used as intermediate inputs. To construct intermediate inputs, we thus concatenate a copy of the original input—the two vectors separated by the [SEP] token—with a second separation token [SEP2] followed by either (i) the last intermediate output, or (ii) all the intermediate outputs so far. Scenario (i), which is illustrated on the right hand side of Figure 3, yields *short intermediate inputs* where the last intermediate output acts as a "pointer" to where

the decoding process stopped in the previous step. Scenario (ii) produces *long intermediate inputs*. While in both scenarios the intermediate outputs need to be appended to produce the final prediction, their prediction routines are slightly different because scenario (i) only needs to concatenate the current intermediate output to the input vectors to produce the next intermediate output, but scenario (ii) needs to append the current intermediate output to the last intermediate input to produce the next intermediate input. Note that there are many other possibilities to construct intermediate inputs (e.g., only including the last two, three, four, etc. intermediate outputs). There are also different options for the intermediate outputs. Instead of being only the next row or pair of tokens, for instance, they could be all the rows or all the token pairs so far.

Table 2: Sentence accuracy achieved by the seq2seq and iterative decoding transformers with relative attention ($r = 8$) on the training set and on multiple test sets of the cartesian product dataset.

| | | Iterative | | | |
| | | short inputs | | long inputs | |
| **Split** | **Seq2seq** | row | token | row | token |
|---|---|---|---|---|---|
| train (up to 5 numbers/letters) | 100% | 100% | 100% | 100% | 100% |
| test (up to 5 numbers/letters) | 97.8% | 100% | 100% | 100% | 100% |
| test (6 numbers, 5 letters) | 14.3% | 89.2% | **100%** | **100%** | **100%** |
| test (5 numbers, 6 letters) | 12.2% | 0% | 99.5% | 0% | **100%** |
| test (6 numbers/letters) | 1.1% | 0% | 98.7% | 0% | **100%** |

To analyze the compositional generalization ability of seq2seq and iterative decoding transformers on the cartesian product dataset, we consider the following experimental setup. Both transformers are trained on samples with up to five numbers and letters. Then, they are tested on the four different test sets described in Section 2.3.2: up to five numbers and letters; six numbers and five letters; five numbers and six letters; and six numbers and letters. The second, third and fourth test sets can be seen productivity tests. Additionally, we only report results for transformers with relative attention (relative radius $r = 8$) as they were the best performing architecture in our experiments.

The average training and test accuracies achieved by the seq2seq model, as well as by the iterative decoding model in the short/long intermediate input and row/token scenarios, are reported in Table 2. We observe that the seq2seq model (first column) achieves 100% accuracy on the training set and close to that on the "easy" test set with up to five numbers and letters. However, it pretty much fails in all of the "hard" test sets, implying that it cannot generalize compositionally to even one extra token being added to the input. The iterative decoding models that decode one row at a time (second and fourth columns) do slightly better as they achieve accuracy closer to the training accuracy in the test set with six numbers and five letters. However, they still fail at the test sets with six letters, which means that the iterative decoding transformer trained to decode one row at a time only generalizes well to calculating cartesian products with a larger number of numbers or, equivalently, of rows.

In contrast, the iterative decoding models that decode one pair of tokens at a time (third and fifth columns) achieve close to or exactly 100% accuracy in all of the compositionally hard splits. This means that the iterative decoding transformer can only generalize to longer iterations when these iterations are the ones that were unrolled during training via iterative decoding. If we take a transformer that has been trained to decode rows, and add one more letter to the input—resulting in one more pair of tokens in each row— it will not be able to predict this additional token pair because it has learned how to unroll more rows through iterative decoding but not more token pairs within a row. We thus conclude that in the cartesian product dataset transformers struggle to learn iteration by themselves, i.e., without the help of iterative decoding. This is an important result because, despite being universal function approximators in theory (Yun et al., 2019), it sheds light onto what transformers can actually learn in practice. Finally, the difference between long and short inputs is not substantial, but longer inputs seem to be better, probably because they provide the transformer with more memory (i.e., more information about the previous intermediate steps).

## 4.3 CFQ

Both PCFG and cartesian product were well adapted for iterative decoding because, in PCFG, the recursive structure of the inputs makes it easy to define the intermediate steps, and in cartesian prod-

uct we have the flexibility to choose their granularity. On the CFQ dataset, defining the intermediate steps is less obvious. The natural choice is to define each intermediate output as a clause of the query as illustrated on the bottom of Figure 3. However, unlike in PCFG—where the order of the intermediate steps was defined by the recursion—and in cartesian product—where the order of the tokens in the input determines the order of the intermediate steps— this ordering of the intermediate steps is not very "natural" because the clauses are sorted alphabetically. Hence, on CFQ transformers also have to learn how to sort.

To make learning this ordering easier for the transformer, we define long intermediate inputs for the intermediate steps. These intermediate inputs are constructed by concatenating the question with all of the previous intermediate outputs so far. As such, on CFQ the iterative decoding prediction routine is the same as for cartesian product with long inputs: we append the current intermediate output to the previous intermediate input to obtain the next intermediate input, and, once the end-of-iteration token is predicted, concatenate all of the intermediate outputs to obtain the query prediction.

Table 3: Sentence accuracy achieved by the seq2seq and iterative decoding transformers with relative attention ($r = 8$) on the training and test sets of the MCD1 split of the CFQ dataset.

| Split | Seq2seq | Iterative |
|-------|---------|-----------|
| train | 99.8% | 99.7% |
| test | 37.1% | 32.5% |

Following this iterative decoding scheme, we compare the compositional generalization abilities of a seq2seq and an iterative decoding transformer on the MCD1 split of the CFQ dataset. The average training and test accuracies are shown in Table 3, where we only report results for the best performing model in our experiments—relative attention with relative radius $r = 8$. Both the seq2seq and the iterative decoding transformer exhibit low compositionality, however, iterative decoding performs worse than seq2seq. This reinforces the limitations of iterative decoding that we observed in cartesian product, namely, that iterative decoding performance is largely dependent on how we define the intermediate steps. In the case of CFQ specifically, we also hypothesize that the worse performance of iterative decoding can be tied to the alphabetical ordering of the clauses, as it does not follow naturally from the grammatical structure of the input. Even though sorting these clauses is something that both the seq2seq and the iterative decoding transformer have to learn how to do, in iterative decoding the transformer has to sort at all intermediate steps, so there are more opportunities to make mistakes. In other words, the error probability is larger in iterative decoding, because it compounds with each intermediate step.

## 5 CONCLUSIONS

This paper introduces iterative decoding, an alternative to seq2seq learning that consists of predicting a series of intermediate outputs from an input, and then adapting these outputs into intermediate inputs that are fed back to the model until a sequence containing an end-of-iteration token is predicted. Through numerical experiments on the PCFG and cartesian product datasets, we demonstrate that, in general, seq2seq transformers do not learn iterations that are not unrolled. By unrolling them, iterative decoding improves transformer compositional generalization. Additional experiments on the CFQ dataset illustrate a limitation of iterative decoding, which is that it depends on how the intermediate steps are defined. In particular, we hypothesize that their ordering is the reason why the seq2seq transformer outperforms iterative decoding on CFQ, as any ordering that is not inherent to the data requires transformers to learn how to sort and iterative decoding may increase the overall sorting error probability.

As part of our future work, we aim to apply iterative decoding strategies to more datasets and understand whether some of the iterative steps can be traded for transformer depth. We also intend to use iterative decoding to investigate the aspects of compositional generalization that transformers can and cannot learn. A clear next step is understanding the effect of the order of the intermediate steps in iterative decoding, and what that effect has to say about the transformer's ability to sort.

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

## A    APPENDIX A: IMPLEMENTATION DETAILS

Across all experiments, the transformer parameters were the same as in the original implementation in (Vaswani et al., 2017), including the learning rate schedule. All experiments were run on machines with a single CPU and four Tesla V100 GPUs with batch size 64 per device.

Table 4: Vocabulary size, training and test samples, and number of training steps for all seq2seq and iterative decoding datasets.

|  | **Seq2seq** | | | **Iterative** | | | |
|---|---|---|---|---|---|---|---|
|  | vocabulary | train | test | vocabulary | train | test | steps |
| PCFG-i.i.d. | 534 | 82662 | 9721 | 535 | 426558 | 9721 | 33325 |
| PCFG-prod. | 534 | 81010 | 11333 | 535 | 346222 | 11333 | 27049 |
| PCFG-syst. | 534 | 82168 | 10175 | 535 | 403808 | 10175 | 31458 |
| Cartesian-row | 26 | 200000 | 1024 | 28 | 600036 | 1024 | 4688 |
| Cartesian-token | 26 | 200000 | 1024 | 28 | 1801869 | 1024 | 14077 |
| CFQ | 181 | 95743 | 11968 | 186 | 682470 | 11968 | 53318 |

For each dataset, and for both the seq2seq and iterative decoding splits, the vocabulary size, the size of the training and test sets, and the total number of training steps is shown in Table 4. The iterative decoding vocabularies are larger due to the addition of special start, end and separation tokens. The number of training samples is larger for the iterative decoding splits because they include all intermediate steps. To make for a fair comparison, the number of training steps is the same for iterative decoding and seq2seq.

## B    APPENDIX B: ADDITIONAL ITERATIVE DECODING RESULTS FOR PCFG

To assess the advantages of iterative decoding under no other sources of compositional generalization, we consider the base transformer (i.e., without relative attention and copy decoder) and analyze its performance on PCFG per number of string editing operations. Namely, in Figure 4 we plot the number of correct predictions achieved by seq2seq (orange) and iterative decoding (blue) on the productivity and systematicity splits of PCFG. We observe that, in the productivity split, the performance improvement comes mostly from samples with a small number of string editing instructions. Consistent with Table 1, without any other form of compositional generalization bias iterative decoding is more helpful with systematicity.

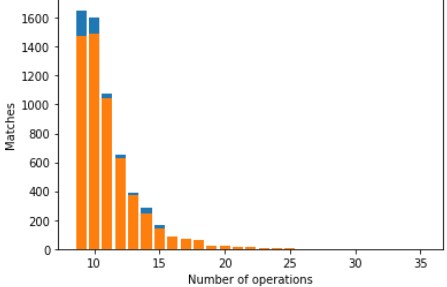 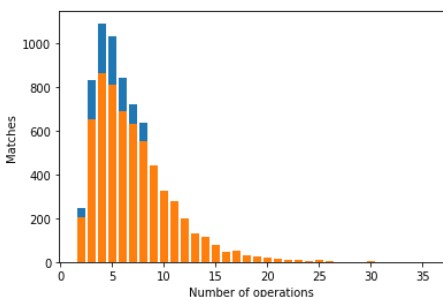

Figure 4: Number of correct predictions versus number of string operations in the input for seq2seq (orange) and iterative decoding (blue), on the productivity (left) and systematicity (right) splits of PCFG.

We draw a similar conclusion from Figure 5, which plots the error per intermediate step ((test error)$^{1/N}$, where $N$ is the number of operations) versus the number of operations in the input for both splits. The error per intermediate step can be seen as the probability of making a mistake at any given intermediate step. On the left, this error approaches one for a smaller number of operations than on the right, indicating that errors compound faster in the productivity split. Interestingly,

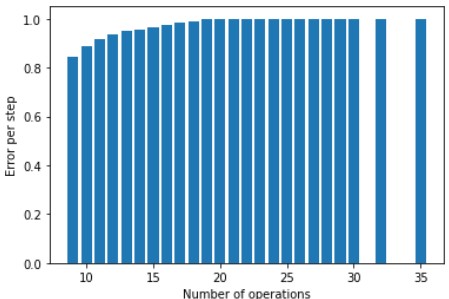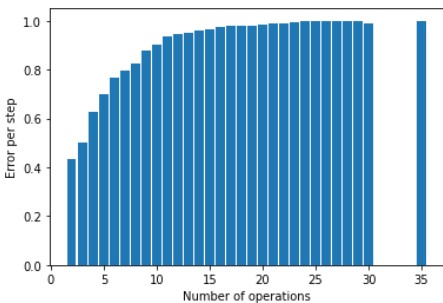

Figure 5: Error per intermediate step versus number of string operations in the input for the iterative decoding transformer on the productivity (left) and systematicity (right) splits of PCFG.

this figure also corroborates our claim from Section 4.1 that decomposing complex instructions into individual operations only helps if the model can execute each operation correctly. In other words, composing individual operations into complex instructions is only one facet of compositionality, but one with which iterative decoding helps.

