# OpenReview forum: "Iterative Decoding for Compositional Generalization in Transformers"
_ICLR.cc/2022/Conference — ICLR 2022 Submitted_

### Official Review · Reviewer_YC1Z · 2021-10-24

**Correctness:** 3
**Technical Novelty And Significance:** 2
**Empirical Novelty And Significance:** 1
**Recommendation:** 3
**Confidence:** 4

**Main Review:**

Strengths:
- The problem studied in the paper, i.e. addressing the inability of Transformers (and more generally, neural models) to generalize compositionally, is an important one.
- The paper is written well.

Weaknesses:
- The biggest weakness of the paper is that it requires the problem to be broken down into iterative subproblems, and it is not at all clear how one might do this with non-synthetic tasks. In fact, sequence-to-sequence is interesting/powerful/general precisely **because** we don't need to manually break down the problem into subproblems.
- I could see some value in this approach if some heuristically-driven approach to creating intermediate input-output pairs resulted in some gains, but this seems clearly not to be the case as evidenced by the underperformance of the proposed approach compared to a regular seq2seq baseline in the CFQ task.

I apologize for the somewhat harsh review, but I truly do not see how this approach would be at all applicable to general tasks. (In fact, it is not at all clear how how one might adapt this approach for even slightly more complex synthetic tasks such as SCAN/COGS.) I am happy to be convinced otherwise though during the discussion (e.g. if this approach is able to meaningfully improve performance on real world low resource sequence-to-sequence learning tasks, I would revise my score upward significantly).

**Summary Of The Paper:**

This paper studies compositional generalization in the context of Transformer models. Noting that "one-shot" sequence-to-sequence learning with Transformer models often results in models that do not generalize compositionally, the authors propose an iterative scheme whereby intermediate input-output pairs are constructed for each task (how this is constructed varies depends on the task), and the decoding proceeds as a series of seq2seq tasks. The approach is found to perform well on synthetic tasks (PCFG, Cartesian product) where there is a "natural" way to construct the intermediate input-output pairs, but fails to perform well on more real-world tasks such as CFQ.

**Summary Of The Review:**

While the approach does improve compositional generalization in certain synthetic tasks, it is almost completely inapplicable to real, interesting problems such as translation or semantic parsing.

---

### Official Review · Reviewer_BXC7 · 2021-11-03

**Correctness:** 3
**Technical Novelty And Significance:** 3
**Empirical Novelty And Significance:** 2
**Recommendation:** 5
**Confidence:** 5

**Main Review:**

Strengths

- The proposed technique is very simple and requires little or no architecture changes to the transformer. It seems particularly effective on PCFG.
- Compositional generalization is a very important problem for the ML community, and the paper presents a useful technique to address this.

Weaknesses

- Some of the results are confusing. PCFG seems like an excellent fit for the proposed approach, but the technique applied to Cartesian Product and CFQ is somewhat strange.
- Given that PCFG was the task that is primarily relevant, the paper would greatly benefit from more detailed error analysis. In other words, the paper is somewhat lacking in content and results as-is.
- There is clearly a lot of missing related work. Compositionality is not a problem only recently found in the ML community...
- (medium priority) The writing could be greatly improved in terms of organization, frequent minor errors, and missing details. Although this seems easily addressable in a revision.

Comments:

a. “By endowing models with the ability to extrapolate to unseen examples that are more complex, or complex in different ways than seen at training, compositionality acts as an implicit mechanism for data augmentation.”

I am not sure it is fair to call “walk right” more complex than “jump right”. Later, complexity is defined: “Compositional generalization (or compositionality) refers to the ability of a model that has learned to perform a set of basic operations—primitives—to generalize to more complex operations, i.e., operations consisting of compositions of the learned primitives”. By this definition, I can see that “walk right” is more complex than “walk” or “right”, and it is only learnable because similarly complex ops (i.e. “jump right”) are seen during training.

I recommend clarifying the quoted statement. Perhaps something like, “unseen examples involving composition not seen during training.”

b. (low priority) “During testing, the model is tested by asking it to compose a longer number of operations than seen during training. Hence, the model has to be able to generalize compositionally.” Very low priority comment… It is not necessary to anthropomorphize the model so much. Possible re-write (take it or leave it): “The test data contains longer sequences with more operations than seen during training. Hence, the model's performance relies on compositional generalization.”

c. You may be interested in another related work/dataset, ListOps (Nangia and Bowman).

d. You may be interested in other works that describe copy mechanism in seq2seq models: Gu et al. 2016 (Incorporating Copying Mechanism in Sequence-to-Sequence Learning) and Gulcehre et al. 2016 (Pointing the Unknown Words). Also, Vinyals et al. 2017 (Pointer Networks), which says plainly in the abstract: “We show that the learnt models generalize beyond the maximum lengths they were trained on.”

In general, it seems the references used in the paper are mostly from recent work, although many mechanism (halting property as another example) exist in many older work that addressed the same problems.

e. How can one tell if the model is learning “shortcuts”? And how does a shortcut differ from overfitting? For example, is a shortcut simply any behavior different from the desired type of compositionality?

f. It would be helpful to re-arrange the figure so they appear closer to the usage in text. This should be very doable because Figures 1 and 3 contain many subfigures.

g. This description is not particularly informative without reading Keysers et al.: “maximizing its compound divergence and minimizing its atom divergence”. Perhaps remove or explain further?

h. Given that relative attention and copy decoder are critical for good performance, they should really be covered in more detail in the main text.

i. The use of “iterative decoding” in Cartesian Product seems very basic and not really representative of intermediate steps. This is partially addressed in the text, but it still seems strange to group this technique with the one used for PCFG. Also, is it possible this is actually addressing a different problem from compositional generalization? Based on how the seq2seq model is used, the problem with seq2seq might be an inability to incorporate large contexts. Similar comment for CFQ.

j. In general, it would be helpful to have more error analysis. Which types of instances failed to predict correctly? Even for the hard splits, often some of these are predicted correctly.

k. As reviewer YC1Z mentioned, it would be helpful to the reader to provide some ideas of how to apply this technique to other tasks.


**Summary Of The Paper:**

This paper presents a type of “task re-formulation” that re-arranges the labels in task such that they are easier for transformer-based encoder-decoder models. This technique is specifically meant to address compositional generalization, which roughly means to generalize to instances longer than seen during training, with combinations of tokens not seen during training, and so on (further clarified in the paper). The paper presents interesting results on three tasks (PCFG, Cartesian Product, and CFQ). I find the problem of compositional generalization with transformers to be very important for the ML community, and that the results are mostly useful. Nonetheless, there is a substantial amount of missing related work, and the writing could greatly be improved.

**Summary Of The Review:**

I believe this paper is slightly below acceptance threshold because of the listed weaknesses. I may not be convinced in the strength / significance of all the results, but perhaps this would be made more clear after addressing problems in the writing.

---

### Official Review · Reviewer_LPPa · 2021-11-04

**Correctness:** 2
**Technical Novelty And Significance:** 2
**Empirical Novelty And Significance:** Not applicable
**Recommendation:** 3
**Confidence:** 4

**Main Review:**

Strengths:
1. The motivation is clearly presented.
2. They provide detailed instructions on how to construct "intermediate input-intermediate output" pairs.
3. They achieve strong performance on PCFG and Cartesian product datasets.

Weaknesses and concerns
1. The construction of "intermediate input-intermediate output" can be difficult for practical problems. For example, it is nontrivial to define sequential subtasks for CFQ(discussed in the paper), classification or machine translation. Furthermore, the construction itself can be seen as a compositional generalization problem because it must know how to identify primitives and execute composition first.
2. missing baselines, A baseline is a seq2seq model trained with the original data as well as the sealed  "intermediate input-intermediate output" pairs. Sealed means the intermediate output transformed to the same form of the original data. This is trivial for PCFG and Cartesian product given your current preprocessing.  For example, given "swap_first_last repeat copy", you can produce samples with "swap_first_last", "swap_first_last repeat" and "repeat copy".  This is because the construction relies on external knowledge while the current baseline does not see it at all.
3. Relative attention and copy decoders are not well-discussed on the effect to compositional generalization. The paper makes the conclusion from improved performance, but (I believe) even a larger model with the same structure can produce stronger performance (e.g., bert-tiny and bert-large). There are lots of variants of transformers, then why these two techniques matter?
4. Experimental results are not compared with other works in this paper. For PCFG(systematicity split), the paper which proposes the dataset report 72 for Transformer while this paper report 64.8. Recent works use T5 without designed components and achieve >50 for CFQ(MCD1 split). However, this paper uses the original transformer(which is not optimized for seq2seq task) and achieves only <40.


**Summary Of The Paper:**

This paper proposes a decomposed training target for transformers to improve compositional generalization. Specifically, a complex task is decomposed into sequential subtasks by external knowledge to exposure the procedure of composition. Then, they train transformers to reproduce these intermediate steps and perform iterative decoding during prediction. Besides, they adopt relative attention and copy decoders to improve the performance of each step, resulting in overall improvements. They achieve strong performance on PCFG and Cartesian product datasets and report negative results on CFQ dataset.

**Summary Of The Review:**

The iterative decoding proposed in this paper has limited to toy datasets. The conclusion about relative attention and copy decoders is not well-supported.

---

### Official Review · Reviewer_BbzM · 2021-11-04

**Correctness:** 3
**Technical Novelty And Significance:** 2
**Empirical Novelty And Significance:** 2
**Recommendation:** 3
**Confidence:** 4

**Details Of Ethics Concerns:**

No ethics concerns.

**Main Review:**

Strengths:
1. The idea of iterative decoding for compositional generalization is interesting and intuitively makes sense. It encourage the model to learn how the reasoning (generalization) is unfolded.
2. The experimental results on the PCFG and cartesian product datasets is promising and show clear improvements compared to seq2seq counterparts.

Weaknesses:
1. The main method (iterative decoding) is described after spending much space on describing datasets, which is somewhat inconvenient for  readers. Maybe consider introducing the datasets in the experiments section will make the paper easier to follow.
2. According to my understanding, iterative decoding will significantly increases the computational costs (roughly x number of intermediate steps). This could be a significant drawback but is not mentioned in the paper.
3. Iterative decoding can be seen as autoregressive generation in the level of reasoning steps. Therefore, if the iterative decoding model is trained with MLE and teacher forcing (training of iterative decoding is not described in the paper), it will face the exposure bias problem. How much is that problem affecting the performance is needed to be investigated.
4. As shown in the CFQ dataset, the proposed only works in cases where intermediate steps can be clearly defined. However, in most cases it is not the case, therefore is unclear whether the proposed method can generalize to many tasks. Therefore the application of the method can be quite limited.

**Summary Of The Paper:**

This paper introduces iterative decoding to improve the compositional generalization ability of seq2seq models. Iterative decoding predicts a series of intermediate outputs from an input, and then adapting these outputs into intermediate inputs that are fed back to the model until a sequence containing an end-of-iteration token is predicted. This approach explicitly encourage the model to learn to unfold the compositional generalization procedure. Experiments on the PCFG and cartesian product datasets show that iterative decoding can improve the compositional generalization ability of seq2seq models. However, additional experiments also show that iterative decoding may fail in more general datasets that do not have explicit intermediate steps defined.

**Summary Of The Review:**

The proposed method is simple but intuitively make sense. However, the method can be limited and only works in certain scenarios. Also, there are efficiency problems and exposure bias problems that are not mentioned in the paper. Therefore I think the paper is below the bar in its current form. I believe it can be improved by addressing the aforementioned weaknesses.

---

### Decision · Program_Chairs · 2022-01-20

**Decision:**

Reject

**Comment:**

All reviewers raise issues with the proposed method and whether it is a) applicable to non-synthetic tasks/datasets; b) how
the input could be broken down into intermediate subproblems in a principled way and whether this would substantially
make the proposal slower than the vanilla encoder/decoder framework; c) awareness of previous work. It is a same the authors
did not provide a response, however the reviewers have provided useful feedback they could use to improve their submission.